# Electroacoustic Biosensor Systems for Evaluating Antibiotic Action on Microbial Cells

**DOI:** 10.3390/s23146292

**Published:** 2023-07-11

**Authors:** Olga I. Guliy, Boris D. Zaitsev, Irina A. Borodina

**Affiliations:** 1Institute of Biochemistry and Physiology of Plants and Microorganisms—Subdivision of the Federal State Budgetary Research Institution Saratov Federal Scientific Centre of the Russian Academy of Sciences (IBPPM RAS), Saratov 410049, Russia; 2Kotelnikov Institute of Radio Engineering and Electronics, Russian Academy of Sciences, Saratov Branch, Saratov 410019, Russia; zai-boris@yandex.ru (B.D.Z.); borodinaia@yandex.ru (I.A.B.)

**Keywords:** antibiotic resistance/sensitivity, bacterial cells, acoustic biosensors, piezoelectric resonators, acoustic delay lines

## Abstract

Antibiotics are widely used to treat infectious diseases. This leads to the presence of antibiotics and their metabolic products in the ecosystem, especially in aquatic environments. In many countries, the growth of pathogen resistance to antibiotics is considered a threat to national security. Therefore, methods for determining the sensitivity/resistance of bacteria to antimicrobial drugs are important. This review discusses the mechanisms of the formation of antibacterial resistance and the various methods and sensor systems available for analyzing antibiotic effects on bacteria. Particular attention is paid to acoustic biosensors with active immobilized layers and to sensors that analyze antibiotics directly in liquids. It is shown that sensors of the second type allow analysis to be done within a short period, which is important for timely treatment.

## 1. Introduction

The wide prevalence of infectious diseases determines the active use of antimicrobial drugs, among which antibiotics are the most important. The use of antibiotics became a revolution in the treatment of infectious diseases. Subsequently, however, doctors have faced a number of unpleasant consequences, notably bacterial antibiotic resistance (the ability of bacteria to adapt to the effects of antimicrobial drugs). Multidrug-resistant pathogens (those resistant to several groups of antibiotics) are of particular danger.

In many countries, the growth of pathogen resistance to antibiotics is considered a threat to national security. The World Health Organization (WHO) has declared antimicrobial resistance one of the 10 global public health threats facing humanity [1]. Therefore, the WHO has made antimicrobial resistance a high priority, as evidenced by the development of the WHO Global Strategy to Contain Antimicrobial Resistance.

The ever-growing use of antibiotics leads to the presence of antibacterial drugs and their metabolic products in the ecosystem, especially in aquatic environments [2]. For example, residues and metabolic products of antibacterial drugs have been found in food, groundwater, and even drinking water [3]. Owing to transfer through the food chain, antibacterial drugs and their degradation products accumulate in the environment and food products. This causes bacteria to develop antibiotic resistance and, ultimately, adversely affects human health.

A major issue is the determination of the sensitivity/resistance of microbes to antimicrobial drugs. In the microbiological laboratory, it is a lengthy and costly procedure. To date, there have been no methods to predict the clinical effect of antibiotics in the treatment of infections with absolute certainty. Yet, the results of the determination of microbial antibiotic sensitivity serve as a guideline for the selection and correction of antibiotic therapy. A very important aspect is the time range within which one can obtain results for the presence or absence of such sensitivity.

Therefore, the development of new technologies and rapid methods for determining microbial antibiotic susceptibility, including those based on biosensors, is a timely task. This review discusses the prospects offered by sensor-based acoustic analysis methods in dealing with microbial antibiotic susceptibility/resistance.

## 2. Antimicrobials and the Mechanism of Their Action on Bacteria

### 2.1. Antimicrobials

Antimicrobials are medicines used to prevent and treat infections in humans, animals, and plants. This group usually includes antibiotic, antiviral, antifungal, and antiparasitic agents. An antibiotic is a substance of natural, semi-synthetic, or synthetic origin that kills bacteria or inhibits their growth. Antibiotics are chosen according to the type of pathogen. The wide variety of pathogenic agents determines the availability of a large number of antibiotic types. An important milestone and the beginning of the real era of antibiotics was in 1928, when Alexander Fleming discovered a protein, penicillin, with antiseptic properties. In 1939, Howard Florey, Ernst Chain, and Norman Heath entered penicillin into production.

Groups of antibiotics, the classification of which has been improved over many decades, differ depending on their mechanism of action and on their structure and origin. The main classification system is the anatomical therapeutic chemical (ATC) classification system (used since 1975). ATC classification partitions drugs into groups according to their therapeutic action and chemical structure. The main antibacterial groups of drugs are shown in Figure 1 [4].

Depending on the mechanism of action on living pathogens, there are:-Bactericidal groups of medicines. These destroy bacteria by disrupting the synthesis of microbial cell wall components and the structure and functions of membranes. These antibiotics include β-lactams, aminoglycosides, fluoroquinolones, glycopeptides, and others (trimethoprim, metronidazole, rifampicin, etc.);-Bacteriostatic groups of medicines. These inhibit the growth and reproduction of pathogens so that the human immune system is able to cope with infections on its own. Bacteriostatic drugs include macrolides, clindamycin, streptogramins, chloramphenicol, and tetracyclines.

Bacteriostatic drugs are divided into several subgroups depending on the nature of their effect on pathogens:-Drugs that disrupt the synthesis of polymers necessary for the construction of the cell membrane;-Drugs that affect the permeability of the cell membrane. This allows active components to penetrate the cell and gradually destroy it;-Medicines that suppress the synthesis of nucleic acids necessary for the normal functioning of microbes;-Drugs that inhibit the synthesis of proteins in the cell.

According to the spectrum of their antimicrobial activity, antibiotics can be conditionally divided into broad spectrum and narrow spectrum.

### 2.2. Antibiotic Resistance

Antibiotic resistance is the resistance of bacteria to an antibiotic. That is, the antibiotic becomes familiar to the bacterium and, therefore, ineffective for treatment. In 2014, WHO experts published a list of antibiotic-resistant “priority pathogens”, which includes 12 bacteria that pose a threat to human health. Mostly these cause nosocomial infections.

A high-priority group includes multidrug-resistant bacteria such as *Acinetobacter*, *Pseudomonas*, and various *Enterobacter* species *(Klebsiella*, *Escherichia coli*, *Serratia*, and *Proteus*). These bacteria have developed resistance to a wide range of antibiotics, including carbapenems and third-generation cephalosporins.

The spread of multidrug-resistant pathogens of the ESKAPE group (*Enterococcus faecium*, *Staphylococcus aureus*, *Klebsiella pneumoniae*, *Acinetobacter baumannii*, *Pseudomonas aeruginosa*, and *Enterobacter* spp.) is a serious threat to modern healthcare [5]. These pathogens often cause invasive infections. If antibiotic resistance develops, the possibilities of treating such diseases can be seriously limited. In addition, these pathogens can spread in community settings or hospitals. *Salmonella* spp. was added to the list of pathogens under CAESAR surveillance in 2016 [6].

In 2017, the obligate pathogen *Neisseria gonorrhoeae*, which affects reproductive health, was included in this WHO list. This was due to the emergence of strains with multidrug resistance to the recommended treatment drugs and a high risk of developing resistance to third-generation cephalosporins (ceftriaxone, cefixime) and azithromycin. The gram-negative *Neisseria gonorrhoeae* is the causative agent of gonorrhea, a sexually transmitted disease (STD). Numerous epidemics of STDs are still important problems in modern medicine because sexually transmitted infections such as syphilis, gonorrhea, chlamydia, and trichomoniasis cause over one million infections every day.

According to the data presented in [7], antibiotic resistance causes 1.27 million deaths per year worldwide. For example, bacterial infections are the most common cause of sepsis, a leading source of morbidity and mortality in low- and middle-income countries [8,9].

*S. aureus*, as one of the main members of the ESKAPE group, causes a wide range of infectious diseases, from soft tissue infections to severe infectious diseases with a high risk of death, such as endocarditis, meningitis, and sepsis [10].

Methicillin-resistant strains of *S. aureus* are resistant to all β-lactams and cephalosporins and are responsible for 13–74% of all *S. aureus* infections [11]. Cases of resistance to vancomycin, daptomycin, linezolid, and teicoplanin have been described in [12,13].

In aquatic environments, antibiotics are ubiquitous at concentrations ranging from ng/L to µg/L [14]. It is necessary to strengthen the study of their toxicity to marine systems and the chronic toxicity of their mixtures.

Thus, problems exist in monitoring the development of antibiotic sensitivity and increasing the effectiveness of antibiotic therapy. This can be achieved by devising methods for the rapid evaluation of bacterial antibiotic susceptibility.

To address these problems, the WHO developed a global action plan to combat antimicrobial resistance in 2015 [15]. One of its goals is to increase investment in diagnostic tools to detect antibiotic-resistant strains and provide faster treatment.

### 2.3. Mechanisms of Antibacterial Resistance

Bacteria can develop antimicrobial resistance (AMR). They can do so through random mutations or through direct exchange of genetic information [16]. That is, a bacterium lacking a resistance gene can get it from other bacteria and instantly learn to fight the new adverse factor (antibiotic). Using more antibiotics may also increase the chance that resistant microorganisms will emerge. Bacteria have evolved a number of defense mechanisms capable of deactivating antibacterial agents, and this has led to the emergence of multidrug-resistant (MDR) organisms [17].

Bacterial resistance to antibiotics may be congenital or acquired. Congenital resistance is determined by the absence of an antibiotic target in microorganisms, or its inaccessibility owing to the initially low permeability of the cell wall or enzymatic inactivation of the antimicrobial drug. This type of resistance is species-specific [18]. Acquired resistance results from a selection of bacteria upon antibiotic exposure, either through mutations in chromosomal or plasmid DNA or through horizontal transfer of resistance genes via plasmids or transposons [17,19]. Increasing concentrations of antibiotics in the environment may lead to the selection of bacteria resistant to antimicrobial compounds, potentially contributing to the emergence of new resistance determinants [20].

Because most antibiotics introduced in the past two decades are modified derivatives of existing drugs, mutations in the genes coding for the synthesis of inactivation enzymes may impart resistance to bacteria [21].

The main mechanisms of MDR (Figure 2) include modification of the antimicrobial agent target [22]; active removal of antimicrobial drugs from cells (efflux) and disruption of the permeability of the bacterial cell wall [23,24]; and enzymatic degradation or alteration of the structure and properties of antimicrobials [25,26].

Antibiotic resistance may be transmitted through a wide range of mechanisms [27,28,29]. Antibiotics may be inactivated (e.g., β-lactamases break down β-lactams such as penicillin) or transported outside the bacterial cell by auxiliary pumps (e.g., the TetA proteins are responsible for tetracycline efflux).

Enzymatic modification of antibiotics through the transfer of functional groups such as acyl, glycosyl, phosphoryl, or thiol groups makes bacteria resistant to a number of antibiotics, including aminoglycosides and macrolides [30].

Most antibiotics act by specific binding to their targets, but some bacteria have developed multidetoxifying enzymes to inactivate clinically relevant antibiotics such as β-lactams, carbapenems, and aminoglycosides. On the basis of the inactivation mechanism used, these resistant enzymes are mainly divided into hydrolytic and modifying.

Bacterial enzymes play a key part in the emergence of antibiotic resistance. The classification of these enzymes is based on their participation in various biochemical mechanisms: modification of enzymes acting as targets for antibiotics, enzymatic modification of intracellular targets, enzymatic transformation of antibiotics, and implementation of cellular metabolism. The main mechanisms of resistance development are related to the evolution of bacterial enzymes owing to the variability of the genes encoding them.

Tens of thousands of enzymes and their mutants that implement various resistance mechanisms form a new community called the enzystome. The role of bacterial enzymes in the development of bacterial resistance to antibiotics is multifaceted [31].

The bacterial efflux systems, which allow bacteria to survive in the presence of antibiotics, are divided into five classes: the ABC transporters (the ATP-binding cassette), MFS (the Major Facilitator Superfamily), MATE (the Multidrug and Toxic Compound Extrusion), SMR (the Small Multidrug Resistance) and RND (the Resistance Nodulation Division) [32,33]. The most clinically relevant class of efflux systems is the RND. These efflux systems are widespread among bacteria, and their genes are almost always present on the chromosome [34].

Bacterial efflux systems are of great interest because they contribute to developing antibiotic resistance. However, it is known that the genes encoding efflux systems are not newly acquired by horizontal transfer but are part of the core genome [35]. Given that antibacterial drugs began to be actively used less than 100 years ago, it is very clear that the function of efflux systems is not limited to the removal of antibiotics from the cell.

Currently, antibiotic resistance is considered an integral part of the evolutionarily formed adaptive potential of a microbial population to external influences. There exist two antibiotic resistance strategies.

The first is the genetically determined property of bacteria to change the targets of antibiotics, their release from the cell, or their destruction. This determines the ability of bacteria to grow in the presence of antibiotics. The rise in antibiotic-resistant strains is the result of selective pressure from poorly controlled antibiotic use.

The other strategy, antibiotic tolerance (AT), is a result of phenotypic transitions that cause the development of a population in the form of a biofilm or the cessation of division and the transition of cells of a small subpopulation to another phenotype-persister cells, which do not divide at all or divide very slowly [36,37]. The emergence and persistence of antibiotic resistance mutations in persister cells is the cause of the emergence of resistant strains [36,38,39].

Genetic determinants of antibiotic resistance were found in normal and permafrost soils 3 million years of age and even older [40,41]. This allows us to consider antibiotic resistance an evolutionarily developed property of controlling the concentration of antibiotics, which act as factors of intercellular communication at the level of the microbial community and biocenosis. Obviously, bacteria solve the evolutionary problem of combating antibiotic effects by acquiring preexisting resistance determinants. Although de novo mutations play some part in the microbial acquisition of drug resistance, horizontal gene transfer through transduction, transformation, and conjugation is of primary importance in the spread of antibiotic resistance determinants [42].

The resistome, which determines bacterial resistance to antimicrobials, was originally formed during evolution to protect microorganisms from naturally occurring bactericidal compounds. Subsequently, this resistome began to change rapidly as bacteria increasingly began to interact with drugs new to them [40,41]. For example, strains that evolved in a drug-free environment for over 50,000 generations are more susceptible to most antibiotics than their ancestors, with most of the change occurring within the first 2000 generations [40,41]. When these strains were exposed to different drug concentrations, the evolved mutants showed a reduced ability to develop resistance, as compared with their ancestor, i.e., the genetic background affects evolutionary pathways to phenotypic resistance [43]. In a subsequent study, whole genome sequencing showed that resistance was caused by different genetic changes as a result of exposure to different drugs and also was a consequence of different genetic backgrounds [44].

Other studies have shown that as selection increases, the benefit associated with drug-resistance mutations also increases, causing an increase in the frequency of mutations in the population and a decrease in overall genetic diversity. By combining experimental microbiology with whole genome sequencing, researchers have traced evolutionary trajectories to resistance and explored the accumulation of mutations in different environmental contexts. This approach has been used widely to study the evolutionary dynamics resulting from antibiotic combinations [40]. The main antimicrobial groups, the genes responsible for the development of antibiotic resistance, and the types of resistance mechanisms are presented in [45].

Thus, during evolution, bacteria have developed many mechanisms of resistance to antibiotics. Their further study and monitoring of microbial antibiotic resistance is of fundamental importance for the development of new methods and the improvement of existing ones to diagnose antibacterial resistance.

## 3. Methods for Determining Antibacterial Sensitivity

### 3.1. Classic Methods

For successful antibiotic treatment, especially in cases of chronic infection, it is necessary to first determine the degree of sensitivity to antibiotics of the microbes that caused the disease. A measure of the sensitivity of microbes is the minimum concentration of the drug (µg/mL), which inhibits the growth of microbes on nutrient media under standard experimental conditions.

Currently, three standard methods, based on the phenotypic change of bacteria, are used to determine the sensitivity of microbes to antibiotics:-Diffusion of the drug into a solid nutrient medium from paper discs;-Serial dilutions in broth;-Phase-contrast microscopy.

Disk diffusion, being one of the oldest, remains the most common method for evaluating antibiotic susceptibility in bacteriological laboratories. The diffusion method is based on the ability of antibiotics to diffuse into agar and arrest, inhibit, or suppress the growth of the test microbe. The rate of diffusion of antibiotic solutions into agar depends on the chemical nature of the drug and on the composition and pH of the medium [46].

Various modifications of diffusion methods have been developed and applied, including methods using wells, grooves, cylinders, blocks, disks, and tablets. These methods are based on the diffusion of antibiotics into an agarized medium and on the suppression of growth of the test culture. The method is suitable for studying most bacterial pathogens and does not require special equipment [47,48,49].

Serial microdilutions in broth are a phenotypic reference method for determining the sensitivity of microorganisms to antibiotics. Serial dilution methods are based on the direct determination of the MIC. For determination of the MIC, specific concentrations of antibiotics are added to a nutrient medium, which is then inoculated with a microbial culture under study. After incubation, the presence or absence of visible microbial growth is evaluated. Depending on the nature of the nutrient medium used, a distinction is made between serial dilutions in broth and serial dilutions in agar. Depending on the volume of the liquid culture medium used, two serial dilution methods are used: the macro method (in vitro) and the micro method (final volume, 0.2 mL or even smaller). The testing procedure is regulated by ISO 20776-1:2006 [50]. For several objective reasons related to the complexity of sample preparation and analysis methodology, serial microdilutions are rarely used in the daily practice of microbiological laboratories [6,51].

Another classic method for determining the sensitivity of bacteria to antibiotics is based on the observation of live bacteria with a phase-contrast microscope. The results of the minimum inhibitory concentration (MIC) of bacteria under the effect of antibiotics can be obtained in as early as 4 h [52].

The currently used methods for AST are very good. They are presented in [53,54,55] and summarized in Figure 3 [53].

Conventional microbiology continues to use the gold standard for antibiotic susceptibility testing in which a sample is cultured and examined via staining. This process of obtaining information about an infection may be very long in the case of slow-growing organisms such as *Mycobacterium tuberculosis* [56,57]. This can lead to fatal time constraints when prescribing antibiotics.

### 3.2. Automated Methods

The effectiveness of infection control depends on the laboratory’s ability to reliably detect antibiotic resistance and its ability to provide clinicians with reliable and comprehensive information as quickly as possible to improve patient outcomes. The identification of antibiotic resistance in bacteria is sped up by the existing sets of automated approaches to evaluating antibiotic resistance. In some systems, only dilution and incubation are automated and bacterial growth is determined by traditional methods. In other systems, all initial operations are performed manually, and only the reading and recording of the results are automated. There are automation systems with the creation of programs for all operations used in determining the sensitivity of bacteria to antibacterial agents (preparation of the sample and bacterial culture, incubation, and reading and recording of the results) [58].

The first automated device to evaluate the sensitivity of microbes to antibiotics was Autobac-1, introduced as a prototype in 1971 [59]. The fully automated system allows the determination of the sensitivity of 40 bacterial samples to 13 antibiotics simultaneously within a few hours. The system compares well with the standard diffusion test. It hasthe potential for application to other endeavors of the clinical microbiology laboratory, with a comparable saving in time and labor.

In the past 30 years, several high-throughput methods to evaluate the sensitivity of bacteria to antibiotics have been developed and widely implemented and are described in detail in [58]. Four automated in vitro diagnosis (IVD) systems have now been approved by the US Food and Drug Administration (FDA): VITEK2 (bioMérieux, Lyon, France), MicroScanWalkAway (Siemens Healthcare Diagnostics, Tarrytown, NY, USA), BD Phoenix (BD Diagnostics, Franklin Lakes, NJ, USA), and Sensititre ARIS 2X (Trek Diagnostics, West Sussex, UK). These systems include various software designed to simplify workflow and minimize interaction with the technologist, as well as to adapt to different regional and institutional environments. Three of these systems produce fast results (3.5–16 h), while the fourth (Sensititre ARIS 2X) requires more time on average. However, even the so-called express methods; require a standardized microbial inoculum, which involves culturing the sample for 24–48 h [53,58].

New technologies have been developed to reduce long diagnostic times [60,61]. One such method is machine learning (ML), a field of artificial intelligence that focuses on developing algorithms for accurately predicting the outcome variables. The application of ML algorithms to evaluating antibiotic susceptibility has attracted a growing interest in the past 5 years owing to the exponential growth of experimental and clinical data, significant investment in computing power, and algorithm improvements [60].

New methods developed to evaluate antimicrobial activity are usually compared with the results of microbiological tests for antibiotic susceptibility. In general, phenotypic methods directly detect the susceptibility of a given microbe to an agent at specific concentrations. In some cases, such methods measure an antibiotic’s minimum inhibitory concentration (MIC).

An alternative method to evaluate antibiotic resistance in bacteria involves the use of microbial genotype rather than phenotype. Genotypic analysis methods are not only faster than phenotypic methods by bypassing laboratory culturing but also provide insight into the mechanisms that govern bacterial sensitivity to antibiotics, ensure the detection of transmission events, and provide important supporting information such as bacterial strain and virulence factors [62].

Molecular techniques can characterize resistant strains. For example, Martín et al. [1] developed a molecular detection method that generates visible aggregates from DNA amplification products and functionalized magnetic nanoparticles. The amplification and detection procedure takes less than 2 h. The factors underlying aggregation were also investigated, and it was found that the amount of amplified DNA products had a positive effect on aggregation. The presented results form the basis for the future development of an ultrasensitive and low-cost approach for detecting bacterial susceptibility to antibiotics, which can be used in small, point-of-care clinics and in other medical institutions.

### 3.3. Sensor Systems for Antibiotic Susceptibility Analysis

Automated methods based on biosensor systems are also successfully used in antibacterial susceptibility analysis. A biosensor is an analytical tool that can analyze the dynamics of antibiotic–microbe interactions. It consists of a sensitive element (bioreceptor) associated with a physical transducer [63]. Biosensors are classified depending on the type of bioreceptor (enzymes, microorganisms, bacteriophages, DNA, antibodies, tissues, organelles, chemoreceptors, etc.) and on the type of sensor (optical, amperometric, potentiometric, semiconductor, thermometric, photometric, and piezoelectric) [64,65]. The general scheme of a biosensor is shown in Figure 4.

In the first stage, the bioreceptor recognizes a substance-specific to it from a multicomponent mixture. In the second stage, information about the course of the biochemical reaction is converted into an electrochemical or another (optical, acoustic, etc.) signal. This stage, which can be called the stage of coupling of electrode and biochemical reactions, is the key to the operation of the biosensor. At the last stage, the electrical signal from the transducer is converted into the form required for processing.

The ability to analyze and monitor bacterial resistance to antibiotics has been shown by an electro-photon approach [66] and by using electrochemical [16,67,68], optical [69,70,71], and nanomechanical [72] sensor systems. The use of cantilever sensors based on recording changes in resonant frequency with an increase in the mass of bacteria in a nutrient medium, depending on the sensitivity, turned out very promising [73]. A system for reading bacterial resistance or sensitivity to antibiotics by using micromechanical oscillators coated with conventional nutrient layers also turned out promising. The potential of this method was shown by determining the resistance of both laboratory and clinical strains of *E. coli* [74].

It is possible to evaluate the effect of different concentrations of an antibiotic (e.g., erythromycin) on a test culture (*E. coli*) cultivated on a solid agar medium by using a split-ring single-port microwave resonator with a resonant frequency of 1.76 GHz and a cross-section of 5 mm^2^ [75]. The change in the amplitude of the output signal decreased with increasing antibiotic concentration, which indicated a decrease in the growth rate of bacteria. The sensor evaluated the antibiotic sensitivity of bacteria in less than 6 h, with the possibility of automating the measurement process.

The work [45] presents the main achievements of sensor technologies for determining the antibiotic resistance of bacteria, but there is no mention of acoustic sensors for solving this problem. Meanwhile, acoustic sensors, which are highly sensitive to the characteristics of the medium in contact with their surface, are very promising for the analysis of antibiotic resistance of bacteria.

Raman spectroscopy is used widely to determine the sensitivity/resistance of bacteria to various antibiotics. Raman spectroscopy can collect molecular fingerprints of pathogenic bacteria in a label-free and culture-independent manner at single-cell resolution. The method based on Raman spectroscopy combined with machine learning to rapidly and accurately identify pathogenic bacteria and detect their antibiotic resistance was reported in [76,77]. The average accuracy of identification of 12 species of common pathogenic bacteria by the machine learning method was 90.73 ± 9.72%. Antibiotic-sensitive and antibiotic-resistant strains of *Acinetobacter baumannii* isolated from hospital patients were distinguished with 99.92 ± 0.06% accuracy [76].

At the beginning of the 21st century, nanotechnological developments have made it possible to widely use the most powerful method of signal amplification surface-enhanced Raman scattering (SERS) spectroscopy. SERS is an analytical tool that combines the specific molecular analysis of Raman spectroscopy with the amplifying signal of plasmonic nanostructures. The predetermined drug sensitivity profiles of urinary tract infection strains allowed the SERS methodology to obtain complete information on relevant antibiotics in less than 1 h [78,79].

## 4. Acoustic Biosensors to Evaluate Antibiotic Effects on Microbial Cells

Methods of electroacoustic analysis are quite promising and increasingly attract the attention of scientists studying various biological interactions. Acoustic sensor systems allow one to analyze biological objects not only by immobilizing active reagents on the biosensor surface but also directly in the liquid phase. In this case, the analysis is carried out within a short period. The described advantages open up prospects for using electroacoustic analysis to record the effect of antibiotics on microbial cells.

Electroacoustic methods are based on recording biospecific reactions in a liquid suspension or an immobilized layer contacting the surface of a piezoelectric waveguide with a piezoactive acoustic wave. Recently, piezoelectric resonators, or delay lines with a propagating surface or plate acoustic wave, have been widely used to make acoustic biosensors. Such sensors are sensitive to changes in the mechanical or electrical properties of a biological object contacting the surface of the waveguide. Acoustic biosensors are most often made on the basis of piezoelectric materials, such as quartz, lithium niobate, or lithium tantalate because they have high chemical resistance [80].

Acoustic waves are excited in a piezoelectric medium, allowing the creation of a whole family of sensors characterized by high sensitivity, fast analysis, low cost, and small size [80]. Depending on the type of waves used, acoustic sensors are classified into sensors based on bulk acoustic waves (BAWs), surface acoustic waves (SAWs) and acoustic plate modes (APMs) [80,81] (Figure 5).

The first mention of using acoustic sensors to assess the effect of antibacterial drugs on bacteria is found in [82].

BAW sensors are resonators in which an acoustic wave propagates between two sides of a piezoelectric plate. These resonators can be divided into those with a longitudinal electric field [80] and those with a lateral electric field [83,84,85,86].

In SAW sensors, a surface acoustic wave is excited by using an emitting interdigital transducer (IDT), propagates along the surface of a piezoelectric plate, and is converted into an electrical signal by using a receiving IDT. Such a sensor can operate at frequencies ranging from a few MHz to several GHz. Surface waves include Rayleigh waves, waves with shear horizontal polarization (SH), and Love waves [81].

### 4.1. Acoustic Sensors with the Active Immobilized Layers

The speed of measurements with the use of piezoelectric biosensors was the reason for close interest in them to determine the sensitivity of microbial cells to antibacterial drugs. The work [87] reported the monitoring of the state of a bacterial biofilm by using a surface acoustic wave sensor. Biofilms are composed of various bacteria that form an extracellular matrix preventing the diffusion of drugs through them. In this case, conventional antibiotic therapy requires a concentration 500 to 5000 times higher than that used to eliminate infections associated with free bacteria. Figure 6 shows the sensor used to monitor the growth and removal of biofilms based on *E. coli* and *P. aeruginosa*.

The sensor is an Al_2_O_3_ structure with a ZnO piezoelectric layer. At the edges of the structure, there are IDTs for the excitation and reception of SAWs at a frequency of about 400 MHz. The space between the IDTs contains the biofilm under study and two side electrodes to supply DC and AC voltages to produce the bioelectric effect. The amplifier, in whose feedback circuit the sensor is included, is an oscillator, and the frequency of generation depends on the mass of the biofilm. This sensor successfully records both film growth and film removal as a result of a complex treatment. This treatment includes exposure to (1) a low dose of an antibiotic (gentamicin), (2) a constant and alternating electric field to promote the bioelectric effect, and (3) a combined action of the bioelectric effect and gentamicin. The most effective film removal is achieved by the combined action of the bioelectric effect and the antibiotic. The disadvantage of this approach is the need to monitor the film for a long period (approximately 50 h). This leads to the need to monitor the temperature because a slight change in temperature can lead to a significant error.

Prospects for the use of electroacoustic methods to evaluate the sensitivity of microbial cells to antibiotics and their antibacterial activity are demonstrated in [88]. This paper describes a method for recording the mechanical vibrations of bacteria before and after exposure to antibiotics by using a quartz resonator, *E. coli*, and the antibiotics polymyxin B and ampicillin. The resonator was included in a special electronic circuit that allowed recording its phase noise, caused by different types of bacterial movement with different frequencies (Figure 7).

If the phase noise strongly decreased after the addition of an antibiotic, this meant that the bacteria stopped moving owing to the lack of resistance to this antibiotic. Combined with pre- and post-experiment cell imaging and colony-forming unit counts, it was found that within 7–15 min, the antimicrobial susceptibility could be tested. However, recording phase noise with a very low intensity leads to the need to use equipment with a low-noise receiver and with a generator highly stable in frequency and amplitude.

The main disadvantages of the above-described sensors with immobilization of analysis components on the sensor surface include the complexity of the immobilization process itself, the need to clean the sensor surface from the reagents used, and the rather long analysis time.

### 4.2. Acoustic Sensors to Evaluate Antibiotic Effects on Bacteria Directly in Liquid

Analysis of the literature data shows the possibility of developing highly sensitive liquid sensors based on resonators with a lateral electric field, in which the electrodes are located on one side of the piezoelectric plate, and the ultrasonic wave propagates in the space between them [83,84,85,86]. Such resonators respond to changes in both the mechanical parameters of the contacting liquid and changes in its electrical properties.

As mentioned, the action of antibiotics may be due to various factors, including inhibition of cell wall synthesis, inhibition of protein and/or RNA synthesis, DNA replication, and membrane dysfunction. Some antibiotics, which represent a separate class of these compounds, are antimetabolites that act as competitive inhibitors. As a result of the antibiotic effects on microbial cells, there may occur changes in cell morphology, destruction of the cell membrane, changes in the cytoplasmic membrane, and subsequent disturbances in biochemical processes in these cellular structures. As a result, there is an acceleration or deceleration of a certain exchange reaction or a change in the permeability of membranes in relation to specific ions or molecules. This, in turn, can lead to a change in the physical properties of the microbial cell suspension, such as conductivity and viscosity. Similar changes in conductivity and viscosity can be recorded with electroacoustic biosensors.

The prospects for using a biological sensor based on a piezoelectric resonator with a lateral electric field to evaluate the effect of antibiotics (ampicillin and kanamycin) on *E. coli* cells and analyze their antibiotic sensitivity are shown in [89,90]. The general scheme of the described sensor is shown in Figure 8.

The resonator with an operating frequency range of 6–7 MHz was made on the basis of an X-cut lithium niobate plate on the lower side, of which two rectangular electrodes were applied. The area around the electrodes and parts of the electrodes were coated with a special varnish, which damped parasitic Lamb waves [91] and provided a rather high-quality factor of ~630. A liquid container with a volume of ~1 mL was glued to the upper side of the plate. The resonator was connected to an LCR meter, which was used to measure the frequency dependencies of the real and imaginary parts of the electrical impedance near the resonance. An alternating voltage applied to the electrodes excited a bulk acoustic wave, propagating between the electrodes along the normal to the sides of the resonator and repeatedly re-reflected between them. The reflection coefficient of this wave from the upper face depended on the electrical conductivity of the suspension. If the cells were resistant to the added antibiotic, the conductivity of the suspension did not change. In this case, the indicated frequency dependencies of the real and imaginary parts of impedance almost coincided (Figure 8b, left). For antibiotic-sensitive cells, these dependencies differed greatly (Figure 8b, right). These results suggest that the physical parameters of suspensions of sensitive and resistant strains exposed to antibiotics correlate with the presence of a plasmid carrying antibiotic resistance. So, an indicator of the sensitivity of bacteria to an antibiotic was the difference between the values of the real and imaginary parts of the electrical impedance of the sensor before and after the action of the antibiotic on the microbial cells. For research, two *E. coli* strains were used: sensitive and resistant to antibiotics. A significant difference in the sensor parameters for the sensitive and resistant strains of *E. coli* under the action of ampicillin and kanamycin was established. The advantages of this approach are the analysis directly in the liquid phase, the high sensitivity, and the short analysis time (10 min). The obtained results demonstrate the promise of a piezoelectric resonator with a lateral electric field for the analysis of antibiotic sensitivity/resistance of bacteria. By using the sensor described above, a comparative analysis was done of the sensitivity of bacteria to β-lactams [84] and aminoglycosides [92].

On the basis of a resonator with a lateral electric field (LEF), a compact acoustic analyzer was developed that demonstrated the possibility of antibiotics rapid analysis by the example of chloramphenicol (CAP) in an aqueous solution [93]. The analyzing part of the device included a liquid sensor based on a LEF resonator made of a Y-X LiNbO_3_ plate, a digital signal generator, and a control microcontroller that interfaces with a personal computer (Figure 9a).

As an analytical signal, the change in the frequency dependencies of the modulus of the electrical impedance of the resonator before and after antibiotic addition to the bacterial suspension was used. If the bacterial cells are sensitive to the antibiotic, these dependencies are very different (Figure 9b); otherwise, they simply coincide. The analysis time, excluding the time of cell preparation for measurement, did not exceed 4 min. This compact acoustic analyzer can be recommended for evaluating the susceptibility of bacteria to antibiotics. The measurements can be made in mobile laboratories without the use of additional expensive equipment.

As already noted, sensors for analyzing the antibiotic sensitivity of bacteria can be made by using surface and plate acoustic waves in piezoelectric plates. For the analysis of a contacting liquid, the most suitable are waves with shear-horizontal polarization [82] and Love waves [94], which are not accompanied by radiation losses in the liquid. These sensors are ideal for liquid analysis and allow the detection of bio-relevant molecules in water or aqueous buffer solutions with high accuracy [95].

Promising sensors for testing the antimicrobial susceptibility of bacteria allow multiple measurements and are easily cleaned during measurements, which is important in clinical research. An example of such a sensor is an acoustic sensor based on a slot mode in a delay line with a zero-order shear-horizontal plate wave [96]. The main advantage of this sensor is the possibility of non-contact analysis, in which the container with the test suspension is isolated from the surface of the delay line. This design of the sensor allows multiple measurements and cleaning of the liquid container without damaging the delay line [96].

The outward appearance of such a sensor is presented in Figure 10a.

The main element of the device is a delay line based on a Y-X plate of lithium niobate with a thickness of 200 µm. Two interdigital transducers (IDTs) are set on the surface of the plate to excite and receive an acoustic wave of zero order with the shear horizontal polarization (SH_0_) and a central frequency of ~3.5 MHz. A liquid container with a volume of 1.5 mL is located above the waveguide of the delay line between the IDTs. The base of the container is made of a Z-X+30° lithium niobate plate, the (X+30°)-axis of which is parallel to the wave vector of the SH_0_ wave in the delay line. A fixed gap between the surface of the delay line and the bottom of the liquid container is provided by using 8-µm-thick strips of aluminum foil.

With the S-parameter meter, the insertion loss and the phase of the output signal of the device were measured. It was found the presence of pronounced resonant peaks in the frequency dependence of the insertion loss associated with the excitation of the slot mode, which resonated over the width of the container base. The changes in the depth and frequency of the resonant peaks pointed above were used as an analytical signal (Figure 10b). By using ampicillin and *E. coli*, it was shown that the depth and frequency of resonance absorption peaks on the frequency dependence of the insertion loss of the sensor changed when an antibiotic was added to a suspension of microbial cells [97]. This meant that the bacteria were sensitive to the antibiotic (Figure 10b, right). The study of an ampicillin-resistant strain by using this sensor showed that the changes in the depth and frequency of the resonant peaks after exposure to the antibiotic were insignificant (Figure 10b, left). Therefore, the obtained results demonstrate the promise of such an approach to analyzing the sensitivity/resistance of microbial cells to ampicillin.

## 5. Conclusions

The most serious challenge that the microbial world poses to mankind is the resistance of pathogenic microflora to the main groups of antibiotics. The fight against antibiotic resistance currently has two priority areas. One consists of the search for new compounds that could make up for the significantly reduced arsenal of drugs. New antibiotics should overcome the resistance of pathogenic microorganisms and not fall under the influence of the resistome already formed in bacteria. The other priority area is to develop ways to “turn off” resistance to known antibiotics. To do so, one has to carefully analyze the mechanisms of transmission, storage, and implementation of resistance to antimicrobial agents [98,99].

One of the goals of combating antibiotic resistance, according to WHO recommendations, is to increase investment in diagnostic tools to detect antibiotic-resistant strains and rapidly prescribe treatment [15]. Therefore, the attention of scientists is drawn to the development of methods for evaluating antibiotic effects on bacteria, allowing the results to be obtained within a short time.

As already mentioned, standard routine methods require at least 48 h to determine bacterial antibiotic sensitivity and select the optimal antibiotic for the treatment of a specific infection. Such a long-term analysis increases the patient’s chances for aggravation of the course of the disease and the development of complications. Therefore, directions for new diagnostic methods are being developed. Undoubtedly, the “gold standard” methods are a guideline in the development of new methods. In addition, because of the increase in antibiotic-resistant strains, the choice of antimicrobials cannot rely solely on the literature data on bacterial susceptibility/resistance. It is necessary to analyze the antimicrobial resistance of bacteria for each culture under study. The results of the sensitivity determination should be a guideline for the selection and correction of antibiotic therapy.

The main challenge in analyzing the resistance/sensitivity of bacteria to antibiotics is to obtain reliable results within minutes or hours instead of days. This review has shown that acoustic biosensors are an alternative method with a promising prognosis for the coming years. The main advantage of acoustic biosensors is that they allow the analysis of bacterial resistance/sensitivity to be done directly in the liquid phase without any use of immobilized reagents. This allows us to greatly reduce the analysis time and to study a large number of samples within a limited time period. However, in spite of the acute market demands, commercially available acoustic biosensors are still under development. Therefore, research on acoustic biosensors may lead to the development of rapid devices for antibacterial drug analysis and medical care in resource-limited settings without any need for trained personnel.

## Figures and Tables

**Figure 1 sensors-23-06292-f001:**
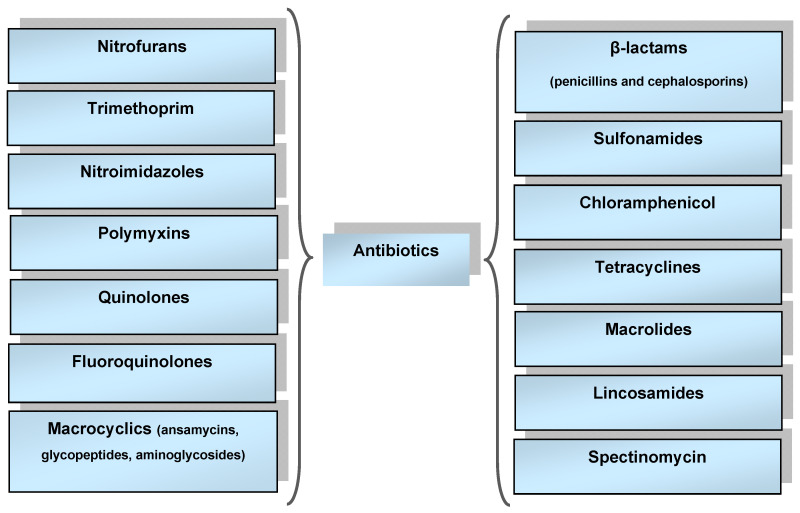
Main groups of antibacterial drugs.

**Figure 2 sensors-23-06292-f002:**
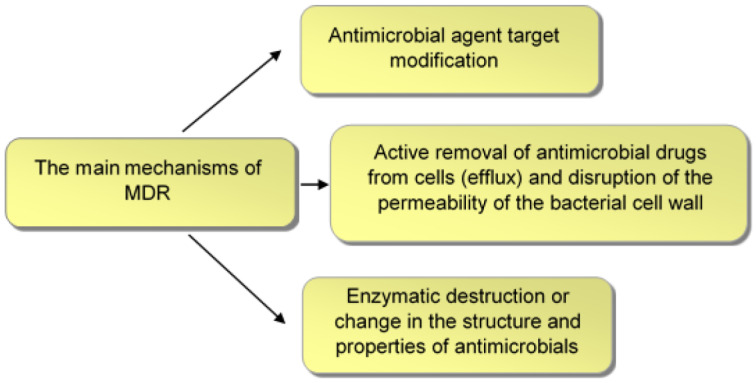
Main mechanisms of bacterial MDR.

**Figure 3 sensors-23-06292-f003:**
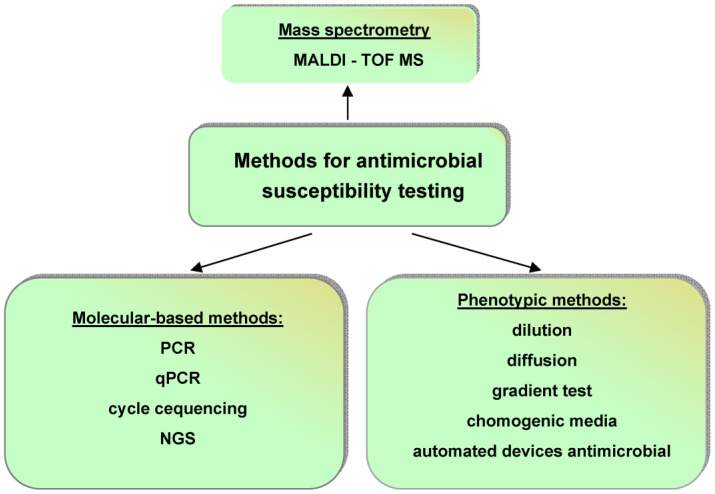
Current methods for antimicrobial susceptibility testing: PCR—polymerase chain reaction; qPCR—quantitative polymerase chain reaction; NGS—next-generation sequencing; MALDI-TOF MS—matrix-assisted laser desorption/ionization time-of-flight mass spectrometry.

**Figure 4 sensors-23-06292-f004:**
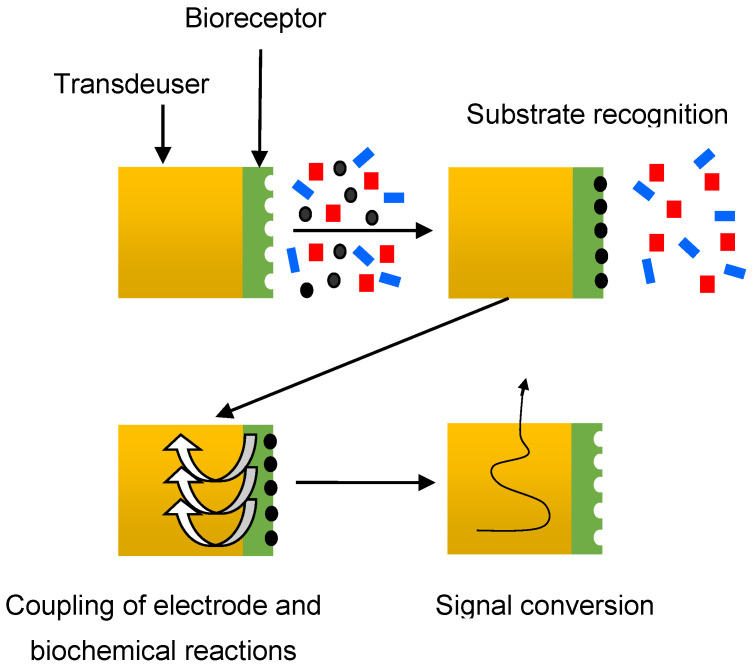
Schematic diagram of a biosensor.

**Figure 5 sensors-23-06292-f005:**
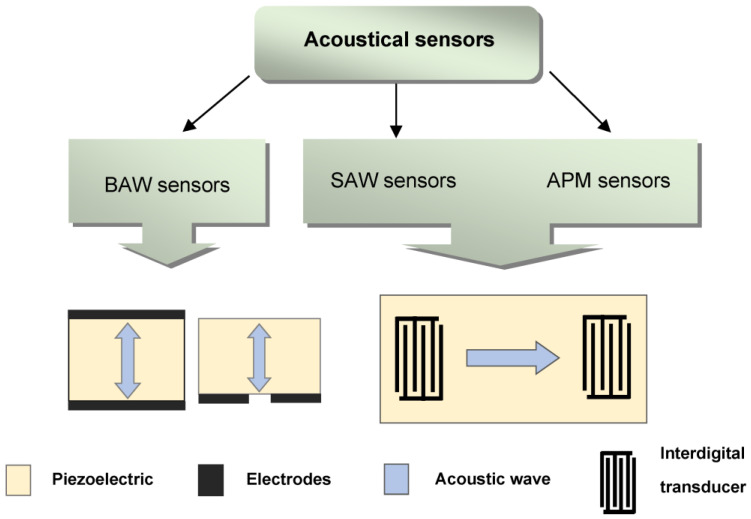
Classification of acoustic sensors.

**Figure 6 sensors-23-06292-f006:**
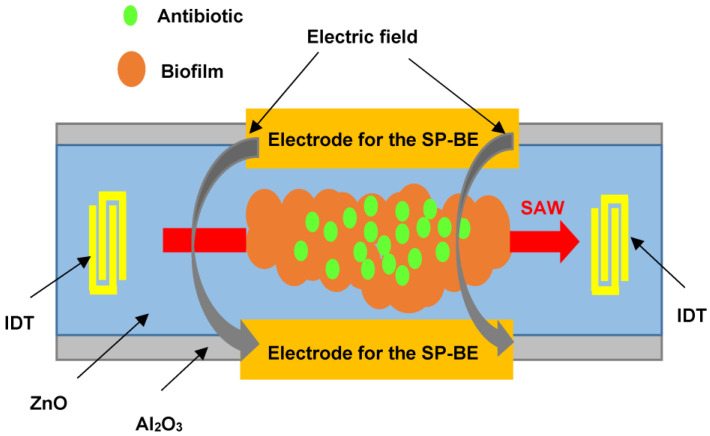
Schematic representation of a SAW sensor for monitoring the state of a biofilm deposited on the sensor surface after exposure to an antibiotic.

**Figure 7 sensors-23-06292-f007:**
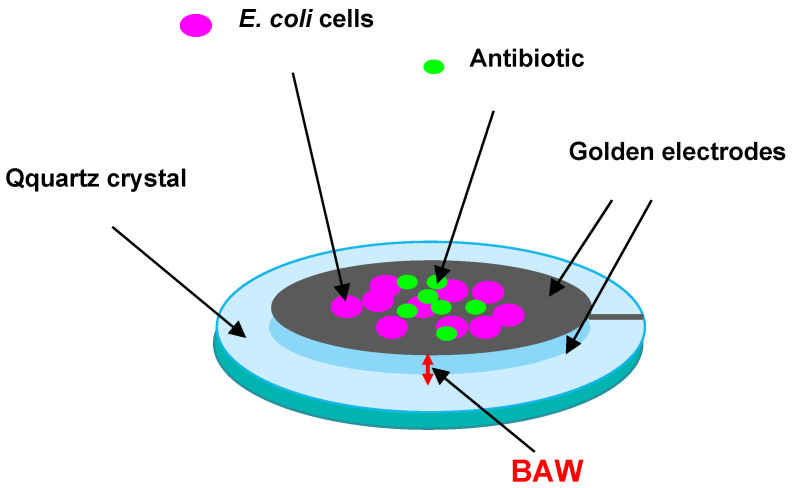
Schematic representation of a quartz resonator with *E. coli* cells deposited on its surface to evaluate antibiotic effects on them.

**Figure 8 sensors-23-06292-f008:**
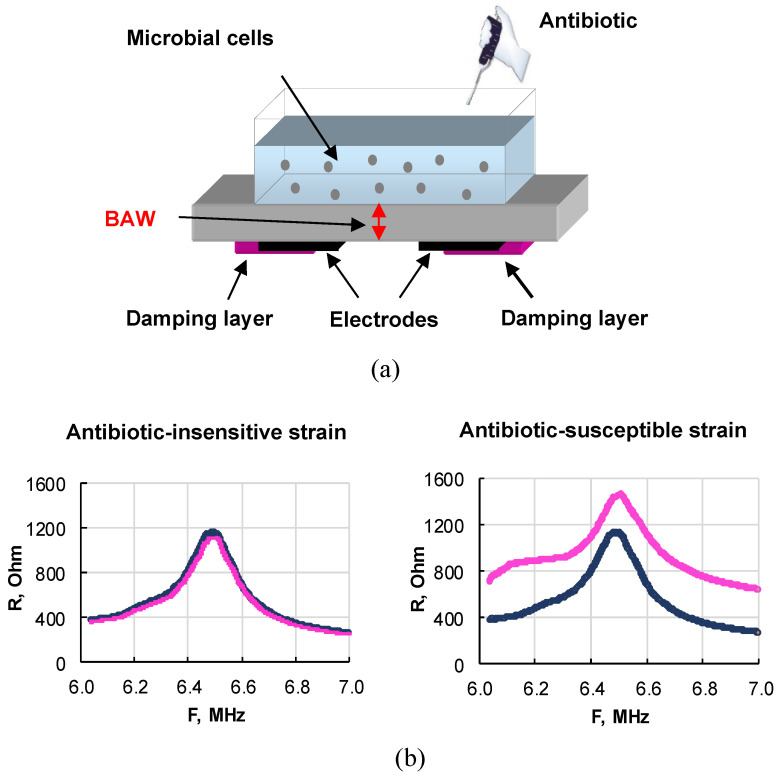
(**a**) General scheme of a sensor based on a resonator with a lateral electric field for evaluating antibiotic-resistant and antibiotic-sensitive microbial strains directly in a liquid. (**b**) Frequency dependencies of the real part of the electrical impedance of the sensor when an antibiotic is added to resistant (**left**) and sensitive (**right**) *E. coli* cells.

**Figure 9 sensors-23-06292-f009:**
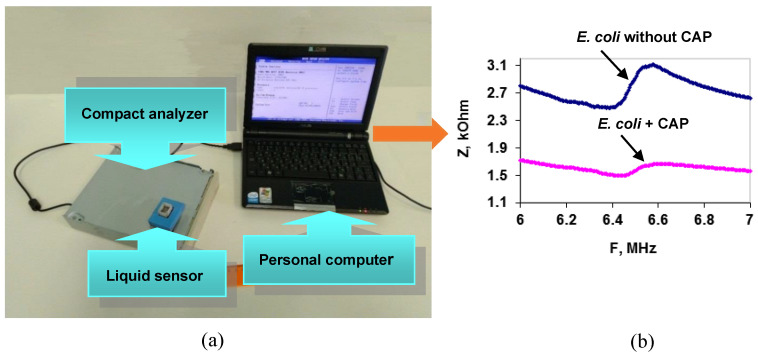
(**a**) Scheme for the compact acoustic analyzer. (**b**) Frequency dependenciesof the sensor’s electrical impedance modulus (Z) for a suspension of *E. coli* cells without (**blue curve**) and after (**pink curve**) CAP addition.

**Figure 10 sensors-23-06292-f010:**
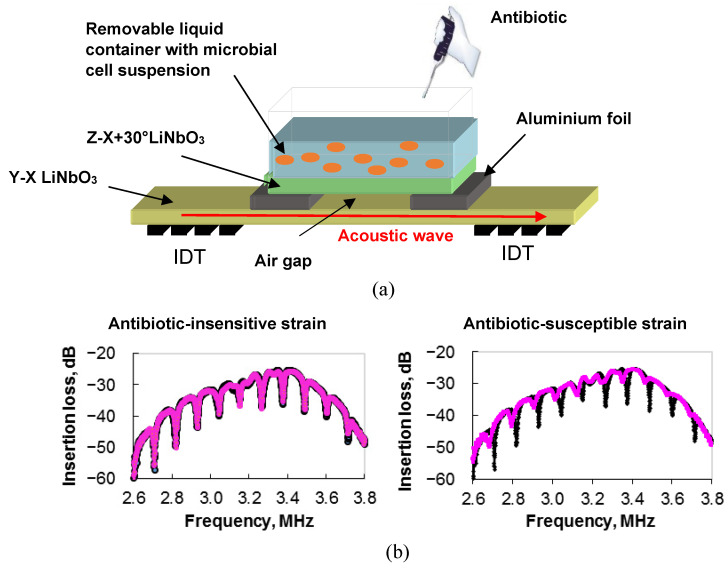
(**a**) The general design of a slot-mode sensor in an acoustic delay line for examining antibiotic-resistant and antibiotic-sensitive microbial strains directly in a liquid. (**b**) Frequency dependencies of the insertion loss of the slot-mode sensor for a suspension of microbial cells insensitive (**left**) and sensitive (**right**) to the antibiotic. Black and pink colors correspond to cell suspension before and after adding antibiotic, respectively.

## Data Availability

Not applicable.

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
