# Peer review of "Electroacoustic Biosensor Systems for Evaluating Antibiotic Action on Microbial Cells"

_sensors, 2023, doi:10.3390/s23146292_

Round 1
Reviewer 1 Report
The manuscript looks like a review type paper. However, it is found to be an article. I am really confused. I would like to consider it as a review paper. As a review paper, the current content and Figures are not enough. So, publication might be considered only if the authors carefully address the following comments (raised by treating the manuscript as a review type paper).
1. The background information about the so-called electroacoustic biological sensor systems and the emergence to write a review are not included at all. The emergence to address the resistance of bacteria is well organized, but this is not enough. The advantages of electroacoustic bio-sensor systems should also be added.
2. For the Figures, if they are reused from published literatures, then proper citation should be added. Otherwise, there would have copyright issues.
3. For the discussion of acoustic bio-sensors and systems, the Figures used in this manuscript are very similar. The advantages of different types of acoustic bio-sensors and systems have not been shown.
4. The Figures and corresponding discussion part should be extended and more details should be included. So that the readers would know more information about the recent progresses and future perspectives of acoustic bio-sensors and systems for assessing of antibiotics action on microbial cells.
5. I suggest the authors to rewrite the Conclusion part, as it now includes only a few contents about the acoustic bio-sensors and systems. According to the title of this review type paper, the summary and perspective should be focused on the acoustic bio-sensors and systems.
There are a few typos and grammar issues in the whole manuscript. Please carefully check and correct.
Reviewer 2 Report
In this manuscript (sensors-2451624), this review concludes the mechanism of antibiotics, antibacterial resistance, and biosensors for analysis of the effect of antibiotics on bacteria, especially the acoustic biosensors. However, there are a quantity of problems needed to be clarified. Thus, this manuscript may be published in Sensors after major revisions.
Some information needs to be clarified:
1. This manuscript is A REVIEW, not original research ARTICLE. Please correct or verify your manuscript formation.
2. I am more interested in the antimicrobial mechanism of the antimicrobial agent, so you should elaborate more on the antimicrobial mechanism of the antimicrobial agent rather than overstating the issue of resistance.
3. Please standardize the format of your references and cite relatively recent references.
4. As far as I know, the combination of Raman spectroscopy and machine learning (ML) has also been studied to distinguish between antimicrobial and non- antimirobial strains, so please review the reference and add to section 3.3.
5. Please check the language expressions in the whole article, there are many sentences I cannot understand.
Please check the language expressions in the whole article, there are many sentences I cannot understand.
Reviewer 3 Report
The review is on electroacoustic biosensors for assessing the effect of antibiotics on microbial cells. The topic of the manuscript is highly relevant and interesting. I have read the whole manuscript with great interest.
However, I found that the review needs to be more structured. Here are some of my suggestions to the authors.
1. From the title, it is clear that the author is trying to focus more on electroacoustic biosensors. However, I find that only a few papers on electroacoustic sensors have been reviewed here. I suggest the author add more works/references in this area.
2. Section 3.1: Describe each method/procedure in a line or two.
3. Some of the works have been passively mentioned here. Instead of mentioning the work, the pros and cons of each model and the author’s opinion about this work should be discussed.
4. Repeatability of the sentence should be avoided.
5. Line 277: briefly describe each method.
6. Line 304/305: add more references.
7. Section 4: more details on different types of sensors (BAW/SAW/APM) should be added.
I strongly recommend the authors resubmit after the corrections.
Some sentences are confusing and hard to follow, so they should be rechecked.
Round 2
Reviewer 1 Report
The authors have addressed the comments. It is now acceptable for publication.
Minor editing of English language required.
Author Response
Please look attached file

Reviewer 2 Report
The revised manuscript (sensors-2451624) has been improved. For the questions about this work, the authors have made careful revisions and supplements and added a numerous relevant and reliable relevance, and the responses are reasonable. And for other written errors, the authors have explained and revised correct. I would like to recommend this paper to be published in Sensors.
Some information needs to be clarified:
1. It is better to give more examples of the novel biosensors of electrochemical in the application of disease detection, toxic detection and other methods applied in the point-of-care, especially in the detection of urine and cells, such as Surface Enhanced Raman Spectroscopy. See some references, Sensors and Actuators: B. Chemical 344 (2021) 130290. Biosensors and Bioelectronics 192 (2021) 113539.Electrochem.Sci.Adv.2022;2:e2100057.
Author Response
Please look attached file.

Reviewer 3 Report
The authors have answered all the questions and made the necessary changes.
Grammer should be checked
Author Response
Please look attached file.
